# CLUE2GEO: FINE-GRAINED IMAGE GEOLOCATION VIA CLUEMAP AND MULTI-STAGE FINE-TUNING

## ABSTRACT

Global image geolocation aims to identify the location where an image was captured, but achieving high precision and robust localization remains challenging.To enhance geolocation precision, we present Clue2Geo, a cue-driven framework for global image geolocation,powered by a Large Vision Language Model (LVLM) for coordinate reasoning. Firstly, an LVLM is employed to extract diverse geographic cues from images, after which the reliability and contribution of these cues are assessed by computing their local consistency and semantic coherence.Based on that,a cue graph named "cluemap" is constructed,which is used as an auxiliary input both during model fine-tuning and inference. Subsequently, we build a large-scale Street View dataset with coordinates and cluemaps to support a three-stage progressive fine-tuning strategy.This strategy is to enhance the downstream model's reasoning capabilities for fine-grained localization tasks.Finally, a post-processing refinement based on Retrieval-Augmented Generation (RAG) using a GPS database is applied after reasoning to reduce the offset of the predicted coordinates, improving both accuracy and stability. Extensive experiments demonstrate that Clue2Geo achieves state-of-the-art performance on fine-grained metrics, particularly at the street levels.[1]

## 1 INTRODUCTION

Global image geolocalization aims to determine the origin of photos taken anywhere on Earth. Compared with traditional region-specific methods(Noh et al., 2017; Cao et al., 2020; Lee et al., 2022; Tan et al., 2021; Shao et al., 2023), it offers broader coverage and more diverse applications. However, achieving accurate and robust global localization remains challenging due to significant variations in landscapes, vegetation, buildings, traffic signs across regions, as well as diverse imaging conditions. High-precision tasks, such as street-level localization, are particularly difficult: models must capture fine-grained local features like building details, road signs, landmarks, etc., which vary subtly across regions and are sensitive to viewpoint, lighting, time, and weather. Existing methods often struggle with sparse or noisy local information, and most datasets focus on city- or region-level localization, limiting support for fine-grained, global-scale geolocalization. These factors together constitute the core technical challenge for high-precision global image geolocalization.

Image geolocalization methods can be broadly categorized into three types: retrieval-based, classification-based, and generation-based. Retrieval-based methods estimate a query image's location by comparing it against a database of geo-tagged images and inferring its position from the most similar matches. While intuitive and effective, these methods involve complex feature extraction, high computational costs for nearest-neighbor search, and a strong dependence on the diversity and completeness of the image database. Classification-based methods partition the geographic space into discrete regions and assign images to the corresponding region. This approach cannot achieve coordinate-level fine-grained localization; the region partitions often fail to reflect true geographic distributions, and the models generally operate as black boxes, lacking interpretable reasoning. Generation-based methods treat geolocalization as a GPS coordinate generation task, using large models to directly predict image coordinates. Although effective on coarse-grained benchmarks, their performance is limited by model knowledge and the inherent randomness of outputs, making high-precision localization challenging.

---

[1]Our code and dataset is available at `https://github.com/xxxxxx`

We propose Clue2Geo, a globally oriented image geolocation framework driven by structured cues, to address the limitations of existing methods. Explicitly extracting and structuring cues can direct models' attention to key features and their relationships.Therefore,geographic cues such as landmarks are extracted from images using an LVLM and organized into a structured cue graph named "cluemap" as an auxiliary input during fine-tuning and inference. Then, an LVLM is employed to predict the coordinates where images were captured.To enhance the model's fine-grained geolocation and reasoning capabilities, a three-stage fine-tuning strategy is applied, using data that integrates cluemaps with the reasoning process to progressively fine-tune the model at different geographic scales and improve its overall performance.Finally, to mitigate the inherent uncertainty,particularly the coordinate deviations in fine-grained localization,a RAG-based post-processing module proposed in prior studies (Zhou et al., 2024; Jia et al., 2024; Vivanco Cepeda et al., 2023) is incorporated. By referencing large-scale image–coordinate pairs to enhance generation, this module refines the final predicted coordinates, significantly enhancing both robustness and accuracy while demonstrating strong generalization capability.We summarize the main contributions of our work as follows:

- We propose Clue2Geo, a framework for global image geolocation that leverages an LVLM to predict coordinates. The LVLM is progressively fine-tuned in stages using structured cues to enhance reasoning and fine-grained localization,and final coordinate is chosen by matching encodings of the fine-tuned LVLM and RAG predictions with image encoding.

- A method for constructing a cluemap is proposed, in which high-value cues are identified by computing their local consistency and semantic cohesion, and other cues are linked to them based on semantic similarity.The cluemap guides LVLM to prioritize informative cues and enhances its ability to understand and reason over complex geographic information.

- A large-scale Mapillary-based dataset containing coordinates, addresses, and visual cues was built and employed for fine-tuning, through which the LVLM's reasoning capabilities and its utilization of complex geographic cues are significantly enhanced.

- Extensive experiments are conducted on two public datasets, IM2GPS3K and YFCC4K. The results demonstrate that Clue2Geo consistently outperforms state-of-the-art baselines in fine-grained localization.

## 2 RELATED WORK

**Image geolocalization** Image geolocalization has received increasing attention in recent years, with applications spanning navigation, urban planning, criminal investigation, and multimedia retrieval. Existing approaches can be broadly categorized into three types: retrieval-based, classification-based, and generation-based methods.Retrieval-based methods estimate the location of a query image by comparing it against a database of geotagged images and inferring its position from the most similar matches (Regmi & Shah, 2019; Shi et al., 2019; 2020; Toker et al., 2021; Zhu et al., 2022; 2021; Workman et al., 2015; Liu & Li, 2019). Vivanco Cepeda et al. (2023) proposed an image-to-GPS retrieval method that explicitly aligns image features with corresponding GPS coordinates to tackle global geolocalization. However, these methods require building and maintaining a reference database, which remains impractical at large scales.To overcome this limitation, Weyand et al. (2016) proposed dividing the Earth into discrete geographic categories for image location prediction. By constructing these categories, predictions can be made at either coarse or fine granularity based on the spatial extent of each category. Nonetheless, classification-based methods (Seo et al., 2018; Vo et al., 2017; Muller-Budack et al., 2018; Pramanick et al., 2022; Clark et al., 2023; Izbicki et al., 2019) often represent each category by its centroid coordinates. Even when a category is correctly predicted, significant errors may occur if the actual image location is distant from the category center.With the rapid development of generative models, generation-based methods have gradually emerged. Zhou et al. (2024) introduced retrieval-augmented generation (RAG) into geolocalization, using coordinates of similar images as references to assist large language models (LLMs) in predicting locations. Building on this, Jia et al. (2024) integrated geographic information into image representations and leveraged geographic diversification and verification to improve prediction performance and robustness. Li et al. (2024) employed LVLMs along with external knowledge from human reasoning to perform geolocalization on street-level imagery.Dufour et al. (2025) applied diffusion models to image geolocalization by treating coordinates as continuous spatial points: they

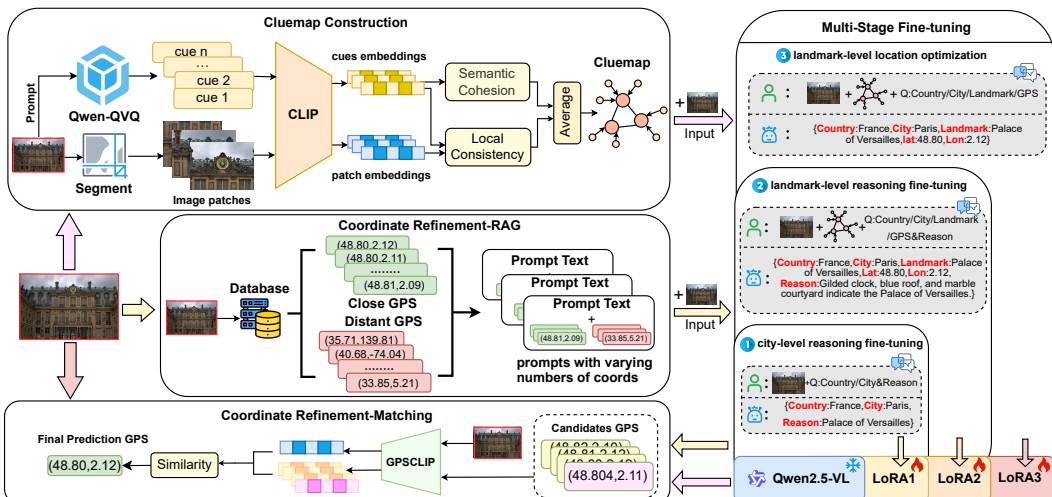

Figure 1: The overall framework of Clue2Geo consists Multi-Stage Fine-tuning, Cluemap Construction, and Coordinate Refinement. A reasoning model is first obtained via three-stage LoRA fine-tuning on Mapillary dataset. Cluemap Construction extracts cues and segments the image, encoding textual cues and patches with CLIP to build a Cluemap as one input, while Coordinate Refinement-RAG retrieves reference coordinates to form another input. At inference, the model processes these two inputs separately to produce candidate coordinates, and the Matching stage uses GPSCLIP to select the final coordinate.

progressively added noise to true coordinates and trained a network to predict the noise conditioned on images, thereby learning the correspondence between visual content and geographic locations.

**Large Multimodal Models** Inspired by the success of large models in single domains, researchers have increasingly focused on cross-modal large models. CLIP(Radford et al., 2021) aligns image and text representations via contrastive learning, achieving strong cross-modal generalization with a simple objective. Building on this, LLAVA(Liu et al., 2023) combines CLIP's visual encoder with the Vicuna language model and employs a two-stage instruction fine-tuning strategy to enhance performance in vision-and-language understanding. In 2023, OpenAI released the multimodal GPT-4V(Achiam et al., 2023), capable of processing both text and image inputs to generate responses, further demonstrating the potential of multimodal models for unified understanding and reasoning tasks. These models not only grasp cross-modal semantics but also exhibit strong capabilities in complex reasoning and generation. Recently, numerous high-performing multimodal models have emerged globally(Jia et al., 2018; 2023; Li et al., 2023). The Qwen-VL(Bai et al., 2023; 2025; Wang et al., 2024) series, as a representative open-source model, combines powerful visual encoding and language understanding, leveraging large-scale image-text pretraining and instruction fine-tuning to achieve superior performance in image-text alignment, cross-modal question answering, and visual reasoning, providing strong support for research in computer vision(Wang et al., 2023), natural language processing(Zhao et al., 2023), and multimodal learning(Yin et al., 2024).

## 3 METHOD

Figure 1 illustrates the overall architecture of Clue2Geo.It contains three components:cluemap construction for extracting and modeling visual cues, three-stage localization fine-tuning to enhance geographic reasoning, and a post-coordinate refinement module to improve predicted coordinates via RAG. The fine-tuning dataset was described at the end.

### 3.1 CLUEMAP CONSTRUCTION

**Visual Cue Extraction** Conventional image geolocation that relies solely on pure image inputs often fails to capture the most distinctive local details essential for fine-grained localization.By explicitly extracting visual cues from images and evaluating their localization value, then combining them

with the original image, the model can focus on the most distinctive and informative regions while retaining overall context, significantly outperforming pure image input in fine-grained localization tasks.Therefore, we first extract geography-related cues from images using Qwen-QVQ.A structured prompt is designed that enumerates seven representative categories of geographically relevant cues, including landmark buildings, natural landscapes, scene types, recognizable text, vehicle characteristics, typical clothing or activities, and overall environmental context. This categorization in the prompt is not a restriction but rather guidance, helping the model form a clearer sense of direction when extracting information. At the same time, the prompt explicitly encourages the model to go beyond these seven categories and capture any additional cues that may aid geolocation, thereby ensuring comprehensive and fine-grained cue collection.

However, the existing reasoning of LVLM is not strictly deductive reasoning, but rather relies on statistical rules, semantic similarity, and retrieval-based memory to "simulate" reasoning, by verifying candidate answers rather than deduction.This verifier-style reasoning can cause cascading errors, where an incorrect initial answer leads all subsequent steps to follow and reinforce the same mistake.Thus, to prevent errors during the extraction stage that could propagate into comprehensive mistakes in subsequent reasoning, the content of the cues is strictly constrained. Specifically, the cues are limited to listing all visually identifiable elements potentially related to geographic location, without making any form of location prediction. In other words, the goal of this step is to maximize the fidelity and completeness of the information rather than directly infer coordinates, thereby providing a reliable, objective, and interpretable input foundation for subsequent geolocation reasoning.The formalization of cue extraction is as follows:

$$C = \{c_1, c_2, \ldots, c_n\} = f_{\text{extract}}(p) \tag{1}$$

$f$ represents an LVLM. Given an input prompt $p$, $f$ extracts a structured set of visual cues $C$ potentially relevant to geographic location, strictly avoiding any form of location prediction.The content of the prompt $p$ can be found in the appendix A.1.1.

**Cluemap Construction** It is worth noting that, due to hallucinations, the cues extracted by LVLM are not always accurate and do not always contribute to geolocation.To assess which cues are accurate and contribute meaningfully, we introduce local consistency and semantic coherence.

Local consistency measures whether the content described by cues is supported by the visual information in the image, that is, whether the cues are accurate and reliable. Since cues often describe local regions of an image, directly computing similarity between a cue and the entire image may not accurately reflect its correctness.We propose two methods of varying complexity to calculate local consistency . Scheme 1 involves performing semantic segmentation on the image, identifying the objects corresponding to the cues, and computing the similarity between them as the local consistency. Scheme 2 divides the image into fixed-size patches, calculates the similarity between the cues and each patch, and takes the maximum value as the local consistency. Since constructing a cluemap for 70,000 images during subsequent fine-tuning would make semantic segmentation computationally expensive and time-consuming, we adopt the simpler Scheme 2 to approximate the local consistency of the cues.The cues and patches are encoded separately using CLIP's(Radford et al., 2021) text and visual encoders, and the cosine similarity is subsequently calculated.The calculation process of local consistency is as follows:

$$S_l(c, I) = \max_{j \in \{1, .., M\}} sim(c, I_j) \tag{2}$$

Here, $c$ denotes a specific extracted cue, $I$ represents the entire image, and $I_j$ denotes the $j$-th image patch after dividing the image into $M$ patches. The function $sim()$ is cosine similarity. The local consistency score $S_l$ of cue $c$ with respect to image $I$ is determined by the maximum similarity between the cue and all patches.

Semantic coherence is defined as measuring the strength of a cue's relationships with other cues, capturing its semantic connectedness within the overall cue set. When multiple cues describe the same object, their coherence scores tend to be higher, which often indicates richer visual elements, and thus potentially greater contribution to geolocation. However, low-coherence "isolated cues" are not assumed to be unimportant; we acknowledge that such cues may still play a critical role. In this work, we retain all cues to form a complete cluemap structure but use semantic coherence solely to describe the strength of relationships without assigning additional weights or special treatment to isolated cues.We employ Sentence-BERT to encode the cues and calculate their pairwise cosine similarity.The calculation of semantic coherence is formulated as follows:

$$S_s(c_i, C) = \frac{1}{|C| - 1} \sum_{\substack{j=1 \\ j \neq i}}^{|C|} sim(c_i, c_j) \tag{3}$$

Let $C = \{c_1, c_2, \ldots, c_{|C|}\}$ denote the set of all cues, where $c_i \in C$ represents the $i$-th cue. The function $sim(c_i, c_j)$ measures the cosine similarity between cue $c_i$ and cue $c_j$. The semantic coherence score $S_s(c_i, C)$ of cue $c_i$ is computed as the average semantic similarity between $c_i$ and all other cues in $C$.

In order to balance both the reliability and the contribution of each cue, the two scores are jointly considered to provide a comprehensive assessment of cue value. Specifically, an equally weighted (0.5 each) average is employed to derive the final relationship strength, which captures not only the semantic connections between a cue and the other cues but also the alignment between the cue description and the actual image. The calculation of relationship strength $S_r$ is formulated as follows:

$$S_r(c_i) = \alpha\, S_l(c_i) + (1 - \alpha)\, S_s(c_i), \quad \text{with } \alpha = 0.5 \tag{4}$$

Finally, based on the final relationship strength, the three highest-scoring cues are selected as primary nodes. The remaining cues are attached to the corresponding primary nodes according to their semantic similarity, thereby forming a graph structure termed Cluemap. In this structure, being a primary node indicates that the cue has comparatively higher and more reliable localization value, while its associated neighboring cues provide complementary information to that primary node. In other words, all cues are organized into three clusters centered on the primary cues, offering a structured representation of the image cues. The constructed Cluemap is subsequently used as supplementary information during fine-tuning and inference to enhance geolocation accuracy.

## 3.2 MULTI-STAGE LOCALIZATION FINE-TUNING

Although LVLMs show strong generalization ability, they struggle with fine-grained geolocation due to insufficient optimization for subtle, domain-specific cues. Fine-tuning on curated data with explicit geographic information adapts the model to these nuances, improving accuracy and interpretability. More importantly, studies have demonstrated that integrating the reasoning process can enhance the capabilities of LLMs (Qiao et al., 2022). Therefore, fine-tuning the model with a focus on reasoning substantially improves its performance in geolocation tasks.

Because geolocation involves hierarchical information ranging from city-level to landmark-level, a single-stage approach is often insufficient. Therefore, we adopt a three-stage fine-tuning strategy: first performing reasoning-oriented fine-tuning at the city and landmark levels, and then conducting location-optimization fine-tuning at the landmark level. This coarse-to-fine adaptation progressively improves the model's reasoning ability and accuracy. Qwen2.5-VL-7B is adopted as the base model for fine-tuning in our framework.

To enhance the model's reasoning capability at the city level, coarse-grained city-level reasoning fine-tuning is first conducted. In this stage, the fine-tuning inputs contain only simple cue reasoning, aiming to train the model to acquire a foundational pattern of coarse-grained geospatial reasoning. An example data template for this stage is shown below (format only; detailed content is provided in the Appendix A.1.2):

> *[Input]: {Image:Image, Question:Which country and city is the image located in?Explain reason.}*
> *[Output]: {Country:"France",City:"Paris",Reason:"French text and the Eiffel Tower."}*

After the model had initially acquired a basic understanding of geographic reasoning, fine-grained landmark-level reasoning fine-tuning is conducted. In the fine-tuning data, the model was explicitly provided with structured cues and was guided to perform step-by-step reasoning, further enhancing its sensitivity to geographic visual cues at the landmark level and its ability to reason from such cues.

The fine-tuning data was built by first constructing cluemaps using the method described in Section 3.1. Geographic visual cues were extracted from images and structured alongside the original images to form multimodal inputs, enabling the model to access both visual information and structured geographic cues. Step-by-step reasoning chains were then generated for each sample using an LLM,

followed by manual review and refinement to ensure the accuracy and logical consistency of the reasoning process. Finally, the image, cluemap, validated reasoning chain, true location, and coordinates were integrated into a complete training instance, and the prompt was designed to explicitly instruct the model to reference the cluemap when reasoning, thereby creating a mapping from visual cues to geographic coordinates. The fine-tuning data template for this stage is as follows:

*[Input]: {Image:Image, Question: Predict the location from the image and clues:ClueMap.Explain reason. }*
*[Output]: {Country:"France",City:"Paris",Landmark:"Eiffel Tower",lat:48.8584,lon:2.2945,*
*Reason:"French text and the Eiffel Tower."}*

In real world geolocation applications, most use cases require only the final coordinate and address rather than a reasoning chain. Therefore, in the final landmark-level location optimization stage, we remove the reasoning outputs and train the model to directly predict precise coordinates and addresses. This design encourages the model to internalize the reasoning process it acquired in the earlier stages and focus on end-to-end geolocation accuracy.The input format from the second stage was retained, with both the original image and the structured cluemap provided so that geographic cues and semantic associations could still be fully leveraged by the model during inference.However, reasoning chains were no longer supplied; instead, the model received the inputs and directly produced the final predicted location and coordinates.This approach strengthened its end-to-end geolocation capability, enabling it to continually align with the real world geographic distribution during training and gradually establish a stable mapping from visual cues to precise coordinates. The fine-tuning data template for this stage is as follows:

*[Input]: {Image:Image, Question: Predict the location from the image and clues:ClueMap.explain reason. }*
*[Output]: {Country:"France",City:"Paris",Landmark:"Eiffel Tower", lat:48.8584,lon:2.2945}*

### 3.3 POST-PROCESSING COORDINATE REFINEMENT

Because of the inherent randomness in LLM outputs, predicted coordinates can inevitably deviate, and repeated predictions for the same image may vary, undermining both stability and accuracy. To address this, we adapt the RAG module proposed in (Jia et al., 2024) as a post-processing step for Clue2Geo, leveraging a retrieval database with large-scale image–coordinate mapping to more precisely align predicted results with real geographic locations and thereby enhance fine-grained geolocation accuracy and robustness.By providing a set of retrieved 'anchor' coordinates, the generative space is constrained to more reliable regions and prediction drift is mitigated. Both the model's direct predictions and the retrieval-augmented outputs are leveraged and compared with the image embeddings to identify the closest match.This process effectively yields multiple independent estimates and chooses the most consistent one, which from a statistical perspective reduces variance and leads to more stable and accurate geolocation results.

The retrieval database consists of a large set of known image–coordinate pairs, with images aligned to coordinates using a pre-trained CLIP model proposed in (Jia et al., 2024)(hereinafter referred to as GPSCLIP). Each image is represented as a high-dimensional feature vector capturing both visual and latent geographic information. The input image is encoded and matched against the database samples based on feature similarity, selecting the most and least similar coordinates as reference. Multiple prompts are constructed based on the number of reference coordinates to separately guide the model's outputs.The outputs, together with the direct output from the fine-tuning stage , are encoded using GPSCLIP, and their similarity to the image embeddings is computed to identify the most similar coordinates as the final prediction. This approach effectively reduces prediction bias and enhances stability and accuracy in fine-grained localization at both city and landmark levels.

### 3.4 MAPILLARY DATASET

We collected 70,000 high-quality street-view images along with corresponding metadata from multiple popular cities worldwide using the open street-view platform Mapillary. Leveraging the latitude and longitude information in the metadata and utilizing Here Maps' reverse geocoding service, we geographically annotated each image and organized locations into three hierarchical levels: country, city, and landmark. Additionally, following the cue extraction method proposed in Section 3.1, we extracted at least five visual cues from each image to support subsequent image-based geolocation reasoning and localization studies.

Table 1: Comparison of Clue2Geo with Other Models on the Im2GPS3k and YFCC4K Dataset.

| Dataset | Method | Street(1km) | City(25km) | Region(200km) | Country(750km) | Continent(2500km) |
|---------|--------|-------------|------------|---------------|----------------|-------------------|
| Im2GPS3k | [L]kNN, $\sigma = 4$ | 7.2 | 19.4 | 26.9 | 38.9 | 55.9 |
| | PlaNet | 8.5 | 24.8 | 34.3 | 48.4 | 64.6 |
| | CPlaNet | 10.2 | 26.5 | 34.6 | 48.6 | 64.6 |
| | ISNs | 10.5 | 28.0 | 36.6 | 49.7 | 66.0 |
| | TransLocator | 11.8 | 31.1 | 46.7 | 58.9 | 80.1 |
| | GeoDecoder | 12.8 | 33.5 | 45.9 | 61.0 | 76.1 |
| | G3 | 14.55 | 37.80 | 52.95 | 70.30 | 83.52 |
| | GeoCLIP | 13.31 | 32.47 | 48.28 | 66.67 | 82.65 |
| | PIGEON | 11.3 | 36.7 | 53.8 | **72.4** | **85.3** |
| | Plonk | 6.17 | 37.17 | 51.28 | 67.00 | 81.85 |
| | **Clue2Geo** | **18.55** | **41.98** | **55.56** | 70.87 | 84.22 |
| YFCC4K | [L]kNN, $\sigma = 4$ | 2.3 | 5.7 | 11 | 23.5 | 42 |
| | PlaNet | 5.6 | 14.3 | 22.2 | 36.4 | 55.8 |
| | CPlaNet | 7.9 | 14.8 | 21.9 | 36.4 | 55.5 |
| | ISNs | 6.5 | 16.2 | 23.8 | 37.4 | 55 |
| | TransLocator | 8.4 | 18.6 | 27.0 | 41.1 | 60.4 |
| | GeoDecoder | 10.3 | 24.4 | 33.9 | 50.0 | 68.7 |
| | G3 | 23.10 | 33.49 | 44.49 | 61.66 | 76.61 |
| | GeoCLIP | 9.59 | 19.29 | 32.61 | 54.98 | 74.67 |
| | PIGEON | 10.4 | 23.7 | 40.6 | 62.2 | 77.7 |
| | Plonk | 6.5 | 32.05 | 43.69 | 59.35 | 75.24 |
| | **Clue2Geo** | **23.39** | **33.77** | **44.66** | **62.46** | **77.78** |

# 4 EXPERIMENTS

## 4.1 EVALUATION DETAILS

The model was fine-tuned according to the strategy described in Section 3.2. Its performance was then evaluated on the public datasets Im2GPS3k (Vo et al., 2017) and YFCC4k (Vo et al., 2017), with all LLM-based methods standardized by replacing their base models with Qwen2.5-VL-7B to eliminate differences arising from model architectures. For each test image, we computed the great-circle distance between the predicted and ground-truth coordinates and quantified the proportion of predictions falling within distance thresholds of 1 km, 25 km, 200 km, 750 km, and 2500 km.

## 4.2 COMPARISON WITH STATE-OF-THE-ART METHODS

To validate the effectiveness of Clue2Geo, we conducted comparative experiments against other state-of-the-art methods on the IM2GPS3K and YFCC4K datasets. Tables 1 present the performance comparison of Clue2Geo with other methods.GeoReasoner (Li et al., 2024) is not included in our comparison, as its results could not be reproduced with the provided weights, its fine-tuning data is unavailable, and the reported metrics were incomplete.

Our approach achieved state-of-the-art performance in fine-grained (street-level) geolocation tasks on both the Im2GPS3K and YFCC4K datasets, while remaining competitive at broader geographic scales. On Im2GPS3K, our method reached 18.55%, 41.98%, and 55.56% within the 1 km, 25 km, and 200 km thresholds, respectively—improving over the second-best method by 4.0%, 4.2%, and 2.6%, with especially notable gains at street and city scales. On YFCC4K, our method achieved 23.39%, 33.77%, and 44.66% on the 1 km, 25 km, and 200 km fine-grained metrics, matching or slightly surpassing the best-performing method (G3) with gains of 0.29%, 0.28%, and 0.17%, while maintaining nearly identical performance on the 750 km and 2500 km coarse-grained metrics (differences of only 0.2–0.4%). These results demonstrate that our approach advances high-precision, fine-grained geolocation without sacrificing accuracy at coarser scales (national/continental level), achieving a stronger balance and greater robustness in multi-scale geolocation tasks.

Table 2: The ablation results of Clue2Geo on Im2GPS3k (with Coordinate Refinement).

| Method | Street(1km) | City(25km) | Region(200km) | Country(750km) | Continent(2500km) |
|--------|-------------|------------|----------------|----------------|---------------------|
| w/o RC | 18.02 | 41.64 | 55.39 | 70.77 | 84.05 |
| w/o RL | 16.35 | 40.71 | 54.21 | 70.20 | 83.88 |
| w/o OL | 15.54 | 38.07 | 52.65 | 69.67 | 83.52 |
| w/o CM | 17.78 | 40.57 | 54.02 | 69.36 | 83.38 |
| w/o CR | 15.45 | 41.04 | 55.26 | 70.47 | 83.92 |
| **Clue2Geo** | **18.55** | **41.98** | **55.56** | **70.87** | **84.22** |

Clue2Geo achieves state-of-the-art performance in geolocation prediction, though its effectiveness varies across different datasets. On the IM2GPS3k dataset, our approach significantly outperforms existing models, primarily because the dataset was originally designed for geolocation tasks. Its carefully curated, high-quality images contain rich visual cues that align closely with the Street View images used for fine-tuning. Street View provides abundant and consistent geographic signals, such as architectural styles and road layouts, enabling the model to effectively learn and transfer fine-grained spatial patterns.In contrast, the YFCC4k dataset, sourced from Flickr, consists of diverse and noisy user-uploaded images. Variations in image quality, camera angles, and lighting conditions, combined with inaccurate location tags and a scarcity of useful geographic cues, create a pronounced domain shift. This limits the model's ability to extract generalizable features, resulting in less pronounced performance gains compared to IM2GPS3k.Despite these challenges, Clue2Geo demonstrates strong performance on both datasets, highlighting its robust generalization ability and effectiveness in handling diverse and noisy data.

Clue2Geo demonstrates significant advantages in fine-grained geolocation tasks, primarily due to high-fidelity cue modeling and multi-stage fine-tuning. By extracting and structurally organizing geographic cues from images, it constructs a cluemap that is both semantically consistent and locally sensitive, enhancing the model's discriminative ability in dense visual feature spaces. The multi-stage fine-tuning strategy strengthens representation transfer from coarse to fine granularity, improving discriminability and robustness at the city and landmark levels, while coordinate refinement leverages real-world geographic distributions to further reduce prediction errors. Through the synergy of these mechanisms, Clue2Geo achieves substantial performance gains over baseline models at fine-grained thresholds, while maintaining superior generalization and interpretability.

## 4.3 ABLATION STUDY

We performed ablation experiments on IM2GPS3K to evaluate our methods.

- w/o RC: Remove the city-level reasoning fine-tuning stage.
- w/o RL: Remove the landmark-level reasoning fine-tuning stage.
- w/o OL: Remove the landmark-level location optimization stage.
- w/o CM: Omit the ClueMap and feed only the raw images.
- w/o CR: Remove the post-coordinate refinement.

Table 2 presents the ablation results on Im2GPS3k. When any individual module is removed, performance drops to varying degrees, and these declines closely match the expected roles of each module. Removing RC lowers Street, City, and Region to 18.02%, 41.64%, and 55.39%, respectively, with the largest drops at street and city levels, confirming its role in building spatial layout awareness and providing a stable foundation for finer-grained reasoning.Removing RL reduces Street accuracy to 16.35% and City to 40.71%, showing that the model's fine-grained reasoning capacity is weakened, underscoring the importance of cues and reasoning processes introduced at this stage for strengthening multi-level reasoning. Removing OL produces the steepest performance drop, with Street, City, and Region metrics falling to 15.54%, 38.07%, and 52.65%, respectively, demonstrating that this module is indispensable for enhancing the model's direct fine-grained location prediction capability; by training intuitive localization beyond reasoning, it preserves both high accuracy and high capacity in multi-scale tasks. Removing CM lowers Street,City and Region accuracy to 17.78%,40.57%

Table 3: The ablation results of Clue2Geo on Im2GPS3k (without Coordinate Refinement).

| Method | Street(1km) | City(25km) | Region(200km) | Country(750km) | Continent(2500km) |
|--------|-------------|------------|---------------|----------------|-------------------|
| w/o RC | 14.58 | 40.44 | 54.35 | 70.14 | 83.52 |
| w/o RL | 9.34 | 36.97 | 53.32 | 70.00 | **84.08** |
| w/o OL | 5.27 | 26.33 | 45.41 | 67.43 | 82.65 |
| w/o CM | 13.61 | 36.87 | 47.81 | 61.83 | 76.24 |
| **Clue2Geo** | **15.45** | **41.04** | **55.26** | **70.47** | 83.92 |

and 54.02%, respectively, indicating that structured cue modeling and multi-level guided reasoning provide valuable contextual constraints that stabilize and improve performance at medium to high granularities. Finally, removing CR leaves City and Region largely unaffected but sharply reduces Street accuracy to 15.45%, highlighting the retrieval module's unique role in mitigating prediction variance, supplying reference coordinates aligned with real-world distributions, and boosting fine-grained stability and precision.

We found that post-coordinate refinement often acts as a "safety net" in the final results, masking the true contributions of other modules. To eliminate the stability compensation and smoothing effects introduced by coordinate refinement, we removed it, allowing the model's multi-scale geolocation and reasoning capabilities to be directly evaluated in a "raw output" state. Table 3 presents the ablation results under this setting.

Without coordinate refinement, removing RC drops accuracy to 14.58% (Street), 40.44% (City), and 54.35% (Region), indicating that RC helps establish coarse spatial layouts and macro-level geographic semantics, thereby laying the foundation for subsequent fine-grained localization. In contrast, removing RL causes a sharp decline to 9.34% (Street) and 36.97% (City), showing that RL mainly supports fine-grained reasoning and multi-cue integration, greatly enhancing the model's ability to recognize complex landmarks and local environments. The removal of OL has the most pronounced effect, reducing accuracy to 5.27% (Street) and 26.33% (City), which underscores OL as a key component for maintaining fine-grained performance. Without OL, the model relies primarily on coarse-grained information, causing fine-level metrics to nearly halve. Similarly, removing CM leads to noticeable drops across all metrics, falling to 13.61% (Street) and 36.87% (City). This demonstrates that CM not only provides essential supplementary information for fine-grained street-level localization but also acts as a stabilizing framework for large-scale reasoning.

These results indicate that Clue2Geo's design is not a simple stacking of modules but a synergistic multi-modal and multi-stage system: the cluemap stabilizes multi-scale reasoning, coarse-grained tuning provides global spatial awareness, landmark-level tuning enhances fine-grained recognition, fine-grained optimization ensures precise direct prediction, and post-coordinate refinement smooths errors and improves robustness. Together, these modules enable the model to achieve breakthroughs in fine-grained geolocation while maintaining coarse-grained accuracy, demonstrating balanced, robust, and high-precision performance across multi-scale tasks.

## 5 CONCLUSION

In this study, we propose Clue2Geo, a global image geolocation framework that integrates LVLMs with structured geographic cues. Unlike previous approaches relying on single visual features or lacking explicit structural modeling, Clue2Geo systematically extracts cues and constructs a cluemap to capture their relationships and importance. By introducing local consistency and semantic coherence, it identifies the most representative cues and models their relationships, providing structured and interpretable inputs for geolocation. We further design a three-stage fine-tuning strategy, from city-level to landmark-level and optimization, to enhance the model's multi-scale reasoning ability. In addition, a post-processing module based on RAG effectively mitigates prediction bias and further improves the stability and accuracy of fine-grained localization. Experimental results show that Clue2Geo achieves significant performance gains across multiple scales and scenarios, demonstrating superior generalization, interpretability, and robustness in cross-domain environments, and offering new directions for global image geolocation research and applications.

## ETHICS STATEMENT

We use publicly available street-view images from Mapillary to fine-tune large models for geolocation tasks.Although Mapillary has already blurred faces and license plates to protect privacy, private residences remain visible in the images. All data usage complies with Mapillary's CC BY-SA license, with proper attribution provided. Users should be aware of potential geographic or cultural biases and avoid applications that could infringe on privacy or enable surveillance. Our resources are intended solely for research purposes.

## REPRODUCIBILITY STATEMENT

Due to space constraints in the main text, we are unable to provide detailed implementation details of Clue2Geo. To ensure reproducibility, we provide the complete implementation details in the appendix, including the cue extraction methods, prompts used for fine-tuning, the construction of the cluemap, and the usage of the retrieval database.

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

# A APPENDIX

## A.1 EXPERIMENTAL DETAILS

### A.1.1 CLUEMAP CONSTRUCTION

For the cue extraction component, we designed the following prompt,as shown in Figure 2.

For the construction of the cluemap, we used **CLIP-cn-vit-h-14** as the visual and text encoder for local consistency , and **text2vec-base-chinese** as the text encoder for semantic coherence. The final importance score was assigned a weight of **0.5**, meaning the importance was calculated as the average of local consistency and semantic coherence.

### A.1.2 MULTI-SCALE LOCALIZATION FINE-TUNING

For the city-level reasoning fine-tuning,landmark-level reasoning fine-tuning, and landmark-level location optimization,we designed the JSON format.The example content is shown in the figure 3, 4 and 5.

Extract and list all visually identifiable cues from the image that may be related to its geographic location:

1) Landmark buildings or distinctive structures — describe their detailed characteristics and quantify them when possible;
2) Natural landscape features (e.g., mountains, coastlines, specific vegetation);
3) Scene type (e.g., urban street view, rural field, indoor);
4) Any recognizable text in the image (e.g., road signs, shop signs, billboards — attempt to transcribe the text content);
5) Types or characteristics of vehicles (e.g., distinctive taxis or buses);
6) Typical clothing or activities of people;
7) Overall environmental atmosphere and style (e.g., weather, lighting).

The above types are for reference only — do not limit yourself to these categories. Each cue should be described in as much detail as possible.
Output format: (cue, detailed description of the cue)

Example:
(Tower structure, A tall tower with a complex metal framework, rectangular base tapering upwards with a pointed top, symmetrical geometric design resembling the Eiffel Tower)

Extract as many cues as possible — there is no upper limit, but provide at least five.
Even subtle cues that might aid geographic inference should be included.
Output only the cues; do not include control characters or any prediction of the geographic location.

Figure 2: Prompt for extracting cues

**"User"**:"According to the content of the picture, please deduce which country the picture appears in. Please give the most likely answer and reasons, output in JSON format, e.g. {'country': ",'city':" 'Reasons':"}.

**"Assitant"**:{'country': 'US', 'city':"Chicago"'Reasons': 'This map shows some of the even numbered US interstates that run east-west. The map also highlights that even numbered US interstate highway numbers gradually increase the further north you travel (I-10 is in the south of the US whilst I-90 in the north). Note'}

Figure 3: The example of city-level reasoning fine-tuning

### A.1.3  POST-PROCESSING COORDINATE REFINEMENT

In the post-processing coordinate refinement stage, for each predicted image we retrieved the 15 most similar and 15 least similar coordinates. Following the procedure described in (Jia et al., 2024), we used three separate prompts, each time providing 5, 10, or 15 of the similar and dissimilar reference coordinates respectively, and generated 1 predicted coordinate per prompt. We then merged these 3 predicted coordinates with the original predicted coordinate to form a set of candidate coordinates. Finally, using the CLIP encoder described in (Jia et al., 2024), we encoded both the image and the candidate coordinates and selected the coordinate with the highest similarity score as the final prediction.

## B  THE USE OF LLMS

We employed LLMs to support our workflow throughout this study. Specifically, we leveraged GPT-4o to assist in debugging portions of the code, helping to identify errors and suggest corrections; and to refine and polish sections of the manuscript, improving clarity, sentence structure, and overall readability.

**"User"**:"Based on the content of the image, infer the country where the scene appears and provide the most likely answer. Then, follow these steps for multi-step reasoning using the extracted clues to identify the most probable city and its landmark:

1. Consider only the most important clue set 1, and determine a few candidate cities and landmarks it may point to.

2. Incorporate the second important clue set 2, and output the candidate cities and landmarks based on combined reasoning from sets 1 and 2.

3. Introduce the third clue set 3, integrate all three clue sets, and output the single city and landmark most likely.

For the predicted location, output the corresponding GPS coordinates.

**Notes:**

- If the city or landmark cannot be determined, fill in `null`.

- Coordinates must be in decimal degrees (DD) format, not degrees-minutes-seconds (DMS) or any other format, and retain at least four decimal places.

- If there is no city or landmark, return the center coordinates of the next higher-level region.

**Input clues in order of importance:**

1. **Clue Set 1:**

    - Symmetrical composition: Four statues symmetrically distributed along a central axis, with equal spacing on both sides, forming a strict geometric layout.

    - Giant stone sculptures: Four male heads carved into the mountain arranged east-west, from west to east: a white male with a wig, a bearded white male, a white male with a dome hat, and a black male with short hair, showing distinct racial features.

    - Coniferous forest vegetation: Tall fir-like trees in the foreground, with conical crowns and straight trunks, approximately 3–5 meters high.

2. **Clue Set 2:**

    - Historical commemorative style: Sculptures in classical realistic style, focusing on facial muscle details, characteristic of late 19th-century public art.

3. **Clue Set 3:**

    - No man-made buildings: Besides the stone sculptures, no houses, roads, or human constructions, preserving the natural mountain shape.

    - Overcast lighting: Sky covered with uniform gray clouds, no direct sunlight, soft mountain shadows, low contrast.

    - Granite mountain: Hard gray-white rock with black spots, visible artificial carving marks, approximately 18 meters high and 24 meters wide.

"Assitant":

- 'country': 'United States', 'city': 'South Dakota', 'landmark': 'Mount Rushmore',
- 'lat': '43.8811', 'lon': '-103.4542',
- 'Reasons': 'Based on the provided visual clues, the location can be inferred as Mount Rushmore in South Dakota, USA: first, the four giant male heads are arranged east-west in a symmetrical composition carved on a granite mountain approximately 18 meters high and 24 meters wide, matching the distinctive features of the Rushmore monument; second, the figures from west to east are Washington (with a wig), Jefferson (bearded), Lincoln (with a dome hat), and Theodore Roosevelt (originally planned as a Native American figure but actually unfinished; the description incorrectly identifies the fourth as a Black male, yet the overall layout points to the four US presidents on Mount Rushmore); additionally, the sculptures are in late 19th-century classical realistic style, emphasizing facial muscle details, consistent with Mount Rushmore's carving style; furthermore, no man-made buildings are visible, the natural mountain shape is preserved, the vegetation consists of tall fir-like coniferous trees, and the lighting is overcast with low contrast, all matching the natural environment of the Black Hills in South Dakota, confirming the scene as Mount Rushmore.'

Figure 4: The example of landmark-level reasoning fine-tuning

**"User"**:"Based on the content of the image, infer the country where the scene appears and provide the most likely answer. Then, follow these steps for multi-step reasoning using the extracted clues to identify the most probable city and its landmark:

1. Consider only the most important clue set 1, and determine a few candidate cities and landmarks it may point to.

2. Incorporate the second important clue set 2, and output the candidate cities and landmarks based on combined reasoning from sets 1 and 2.

3. Introduce the third clue set 3, integrate all three clue sets, and output the single city and landmark most likely.

For the predicted location, output the corresponding GPS coordinates.

**Notes:**

- If the city or landmark cannot be determined, fill in `null`.

- Coordinates must be in decimal degrees (DD) format, not degrees-minutes-seconds (DMS) or any other format, and retain at least four decimal places.

- If there is no city or landmark, return the center coordinates of the next higher-level region.

**Input clues in order of importance:**

1. **Clue Set 1:**

   ○ Traditional architecture: The buildings feature a typical East Asian style, with red pillars and gray tiled roofs, decorated with exquisite painted patterns, and have a symmetrical and solemn structure.

   ○ Scene type: An outdoor courtyard with a stone-paved ground, surrounded by traditional buildings and modern signboards, clean and historically atmospheric.

   ○ Signboard: A vertical black signboard on the left side of the courtyard with white text and patterns, though the exact content is unclear.

2. **Clue Set 2:**

   ○ Stone steps and platform: Several wide stone steps in the center of the square leading to a high platform, with railings on both sides, designed simply and rustic.

   ○ Human activity: Several people walking or standing on the open space in front of the building, wearing modern casual clothes,seemingly visiting or sightseeing

3. **Clue Set 3:**

   ○ Red pillars: The building's supporting pillars are bright red, with smooth surfaces and stone bases at the bottom, making them very striking.

**"Assitant"**:'country': 'South Korea', 'city': 'South Korea', 'landmark': '1 street', 'lat': '37.28178572', 'lon': '127.0131513',

Figure 5: The example of landmark-level location optimizarion

