# OpenReview forum: "Clue2Geo: Fine-Grained Image Geolocation via Cluemap and Multi-Stage Fine-Tuning"
_ICLR.cc/2026/Conference — ICLR 2026 Conference Withdrawn Submission_

### Official Review · Reviewer_Zwz8 · 2025-10-23

**Soundness:** 2
**Presentation:** 1
**Contribution:** 2
**Rating:** 2
**Confidence:** 5

**Summary:**

This paper presents a framework called Clue2Geo for global image geolocation. It contains an LVLM component and a postprocessing refinement based on RAG. Additionally, a street-view dataset is proposed for the LVLM fine-tuning. This dataset provides images, coordinates, addresses, and constructed visual cues (named “cluemap” in this paper). The experiments on the public benchmark IM2GPS3K and YFCC4K show the better performance of Clue2Geo, and ablations show the contribution of different components.

**Strengths:**

1. Novel cluemaps are introduced to supplement the prompt for the LVLM
2. The combination of all components leads to decent performance

**Weaknesses:**

Based on the thorough review of the manuscript, several significant weaknesses have been identified:

- At the bottom of the first page in this manuscript, there is a github repo https://github.com/xxxxxx, which is not an anonymous github. I am not sure if this potentially violates double-blind review principles.
- Besides the above issue, there are also some concerns regarding the Method and Experiments.
  - Method
    - This manuscript claims that  "Since cues often describe local regions of an image" (line 191-192), it is no doubt, however, the authors choose Scheme 2, which divides the image into fixed-size patches, calculates the similarity between the cues and each patch (line 195-197). This approach sounds unreasonable because the small single patch may not be associated with any large-scale buildings. These splits break the structure of the image and introduce bias during the process of evaluating cues.
    - The authors empirically set the $\alpha$=0.5 (line 230-231). I want to know whether different settings of $\alpha$ have an impact on the cluemap construction. Is there more discussion about it?
    - Regarding the landmark levels and some examples mentioned in this manuscript (line 250-278), there is a question here: does every image have a landmark? The example shows the “Eiffel Tower” as the predicted landmark, but how about other images?
  - Experiments
    - The proposed Clue2Geo uses street-view images as training data (line 318), but there is no discussion about its performance on street-view benchmarks like [1] and [2], or a manually curated street-view test set
    - The Clue2Geo contains an LVLM component, but there is no comparison with open/closed-source LVLMs, e.g., LLaVA, Qwen-VL, or GPT4V, as you mentioned in Related Work
    - The Clue2Geo uses the same RAG module proposed in G3 [3], which includes a large-scale retrieval database. It would be better to give more comparisons between these two methods, such as efficiency, failure cases, and intermediate results; otherwise, the proposed Clue2Geo may appear unconvincing, as the performance gains could be attributed primarily to the G3 RAG module rather than the method itself.
    - In Table 1, you should clearly indicate which data you obtained by redeploying the tests yourself. It seems that many of the results are different from those reported in the paper.
    - The Clue2Geo achieves SOTA performance among the methods compared in this manuscript; however, another accepted paper [4] reports results that surpass this SOTA.

- Literature review: This manuscript has some wrong statements about some work
  - Line 063-064, GeoCLIP[5] does not contain a RAG-based post-processing module
  - Line 365-367, GeoReasoner[6] evaluates only on a highly locatable subset of IM2GPS3k[7], making its results incomparable
  - Line 391-400, the IM2GPS3k[7] was also sourced from Flickr and consists of diverse and noisy user-uploaded images, similar to YFCC4k[7]; this cannot be regarded as a meaningful difference between them.

- Writing: This manuscript has many writing issues, such as missing spaces between sentences (line 012, 014, 017, 018, 021, 022, ......), incorrect quotation mark usage (line 173), and does not provide citations or footnotes when first introducing specific platforms, datasets, or model names (line 077, 164, 215, 396)



[1] Where we are and what we’re looking at: Query based worldwide image geo-localization using hierarchies and scenes, CVPR 2023
[2] OpenStreetView-5M: The Many Roads to Global Visual Geolocation, CVPR2024
[3] G3: An Effective and Adaptive Framework for Worldwide Geolocalization Using Large Multi-Modality Models, NeurIPS 2024
[4] GeoRanker: Distance-Aware Ranking for Worldwide Image Geolocalization, NeurIPS 2025
[5] GeoCLIP: Clip-Inspired Alignment between Locations and Images for Effective Worldwide Geo-localization, NeurIPS 2023
[6] GeoReasoner: Geo-localization with Reasoning in Street Views using a Large Vision-Language Model, ICML 2024
[7] Revisiting IM2GPS in the Deep Learning Era, ICCV 2017

**Questions:**

Please see the Weaknesses

---

### Official Review · Reviewer_TX7S · 2025-10-27

**Soundness:** 2
**Presentation:** 2
**Contribution:** 2
**Rating:** 2
**Confidence:** 3

**Summary:**

This paper proposes a new pipeline for global image geo-localization called Clue2Geo. The central idea is to convert LVLM-generated semantics into a “clue map” and use a multi-stage fine-tuning strategy to improve fine-grained localization. Experiments on IM2GPS3K and YFCC4K report state-of-the-art results, with the largest gains on street-level tracks.

**Strengths:**

- Leveraging LVLMs to generate additional semantic cues beyond images is a compelling direction. Formalizing LVLM outputs as a ClueMap and fusing them with image patches in a multi-stage regimen is novel.
- The proposed method achieves state-of-the-art performance on two benchmarks, the improvements are more significant on fine-grained street-level geo-localization.

**Weaknesses:**

- The motivation and the challenge this paper aims to address are not clear. The proposed method reads like a stack of components (ClueMap, multi-stage finetuning, RAG) without a clear motivation and a focus on key challenges. For example, why is a ClueMap needed, and why in combination with multi-stage tuning and RAG, what challenge in previous works does the proposed method solve?
- Eq. 3 introduces a term that calculates the “semantic coherence”, which compares the similarity of a given cue across the set of all cues. The definition and scope of C are ambiguous (if it is for each image or the whole dataset). Also, the underlying mechanism of this semantic coherence is a bit hard to understand. The intuition and rationality under this design need further clarification. Moreover, no ablations or analyses are performed to validate the effectiveness of this strategy.
- In section 3.1, the graph structure is not explained in text or figures, making it hard to imagine what does the graph look like, and how to apply the graph in the pipeline.
- It is unclear how the LVLM-based fine-tuning is performed given the data template, the training loss, optimization settings, and schedules are needed.
- Experiments are performed on small datasets, the generalization on larger datasets like the YFCC26k needs further validation.
- Implementation details about training and evaluation are missing, making it hard to assess the fairness of the comparison and reproduce the results.
- Formatting issues:
1) Missing spaces around punctuation in places
2) The provided github link is not working
3) Inconsistent use of \citep{} and \cite{}

**Questions:**

see weaknesses

---

### Official Review · Reviewer_BZAd · 2025-10-29

**Soundness:** 3
**Presentation:** 3
**Contribution:** 2
**Rating:** 4
**Confidence:** 4

**Summary:**

This paper proposes Clue2Geo, a cue-driven global image geolocalization framework that uses a LVLM for fine-grained coordinate reasoning. It constructs a ClueMap, a structured graph of visual cues extracted from images, where each cue’s reliability is assessed through local consistency and semantic coherence. A three-stage fine-tuning strategy progressively enhances geographic reasoning from city to landmark levels, followed by a retrieval-augmented refinement module to improve coordinate accuracy. Experiments on IM2GPS3K and YFCC4K show that Clue2Geo achieves state-of-the-art fine-grained localization, outperforming previous methods such as GeoCLIP and G3 at street and city scales.

**Strengths:**

1. The paper is logically well-structured, with clear figures and tables, providing a pleasant reading experience.
2. The idea of extracting cues and constructing a cue graph is quite smart, as it introduces a new data structure to the field.
3. The related work section is comprehensive and covers nearly all existing methods in this area.

**Weaknesses:**

1. In many places, a space should be added after the period before continuing the sentence, but it is currently missing.
2. The second paragraph of the Introduction section has no citations at all and should be revised.
3. The open-source link is only a placeholder and does not provide an actual anonymous repository.
4. Typo in line 173, the quotation marks are incorrect.
5. In line 194, the description of Scheme 1 is very unclear and difficult to understand.
6. In Section 3.2, is the training performed stage by stage, that is, after finishing one stage, you attach the LoRA and continue training in the next stage? This is somewhat similar to GeoReasoner, which could be cited here.
7. In Section 3.4, will the dataset be released?
8. The implementation details section is missing.
9. The experimental section is rather limited, containing only overall results and ablation studies. There is no hyperparameter analysis, such as the effect of different backbones or backbone scales on the results.

Overall, this paper is a good work, but the experimental section is not solid enough. The current results are too limited to convincingly demonstrate the effectiveness of so many modules. If the authors can provide more experimental evidence to support the validity of these components, I would be willing to adjust my score.

**Questions:**

please refer to the weakness section

---

### Note · Authors · 2025-12-01

I have read and agree with the venue's withdrawal policy on behalf of myself and my co-authors.